# Iron, Copper, and Zinc Homeostasis: Physiology, Physiopathology, and Nanomediated Applications

**DOI:** 10.3390/nano11112958

**Published:** 2021-11-04

**Authors:** Robert Szabo, Constantin Bodolea, Teodora Mocan

**Affiliations:** 1Physiology Department, “Iuliu Hatieganu” University of Medicine and Pharmacy, 400000 Cluj-Napoca, Romania; robert.szabo@umfcluj.ro (R.S.); teodora.mocan@umfcluj.ro (T.M.); 2Clinical County Emergency Hospital, 400000 Cluj-Napoca, Romania; 3Municipal Clinical Hospital, 400139 Cluj-Napoca, Romania; 4Nanomedicine Department, Regional Institute of Gastroenterology and Hepatology, 400000 Cluj-Napoca, Romania

**Keywords:** nanoparticles, iron physiology, copper physiology, zinc physiology, hepcidin, inflammation, sepsis, nutritional immunity, iron deficiency, iron overload

## Abstract

Understanding of how the human organism functions has preoccupied researchers in medicine for a very long time. While most of the mechanisms are well understood and detailed thoroughly, medicine has yet much to discover. Iron (Fe), Copper (Cu), and Zinc (Zn) are elements on which organisms, ranging from simple bacteria all the way to complex ones such as mammals, rely on these divalent ions. Compounded by the continuously evolving biotechnologies, these ions are still relevant today. This review article aims at recapping the mechanisms involved in Fe, Cu, and Zn homeostasis. By applying the knowledge and expanding on future research areas, this article aims to shine new light of existing illness. Thanks to the expanding field of nanotechnology, genetic disorders such as hemochromatosis and thalassemia can be managed today. Nanoparticles (NPs) improve delivery of ions and confer targeting capabilities, with the potential for use in treatment and diagnosis. Iron deficiency, cancer, and sepsis are persisting major issues. While targeted delivery using Fe NPs can be used as food fortifiers, chemotherapeutic agents against cancer cells and microbes have been developed using both Fe and Cu NPs. A fast and accurate means of diagnosis is a major impacting factor on outcome of patients, especially when critically ill. Good quality imaging and bed side diagnostic tools are possible using NPs, which may positively impact outcome.

## 1. Introduction

Divalent metal ions are part of the physiology of complex cellular organisms such as humans but are also essential for the simpler organisms such as bacteria and fungi. The most abundant ion in humans is iron, a divalent element essential for many processes. Like iron, both copper and zinc are divalent metals possessing the ability to partake in Fenton reaction [1]. Byproducts of reduction and oxidation produce cell damage by denaturing proteins such as enzymes and DNA but also phospholipid structures such as cell membranes. These processes are a double-edged sword because, on one hand, they damage the self-organism, and on the other, these ions can act as defense mechanisms when aimed against foreign organisms. The fine balance is maintained by a complex network of homeostatic mechanisms.

Fe, Cu, and Zn are cofactors in processes of protein synthesis. This explains why their role is essential in the functioning of many organs and systems (Figure 1) [2]. Some proteins act as enzymes while others play a structural role; for example, the hemoglobin and DNA. In simpler organisms such as bacteria, divalent ions are involved in enzyme production, transcription, and signal transduction, which confer them the ability to be virulent and to replicate [3]. The shared need for these elements creates competition between invader (bacteria) and host (human). When withheld by the host, it represents a natural defense against the invader.

In humans, altered levels lead to different disorders [4]. Inadequate intake is often the case and produces absolute deficiencies. In some cases, the inadequate levels are secondary to genetic or acquired diseases, in which disturbed homeostatic processes are caused by absent or nonfunctioning components (Table 1) [5,6]. Due to the severe impact that both deficiencies and an overload of these ions have on the human body, new ways to restore ion homeostasis have been researched. Iron, copper, and zinc nanoparticles possess pleiotropic properties. While the small size provides better penetrability, coatings of the particles confer the ability to target specific cells and tissues. These biological properties combined with the physical properties of iron particles allow for their collective use as a contrast agent for diagnosis and as a therapeutic agent for theranostics. It is essential to understand the mechanisms involved in order to produce and provide personalized treatment. This review article offers an overview of the mechanisms involved in ion homeostasis and ways to alter these pathways, using nanoparticles, to diagnose and treat various disorders.

## 2. Iron Metabolism

### 2.1. Iron Uptake

Iron homeostasis is essential for living and is found in both ferrous (FeII) and ferric (FeIII) forms. At physiological pH, the former is more soluble, with a trade-off being the inclination to take part in reactions that generate hydroxyl radicals. As a result of radicals, lipid peroxidation takes place and causes oxidative stress (ROS), which—if it occurs inside a cell—denatures vital proteins. Although more stable, ferric iron is insoluble. To overcome the insolubility, FeIII in humans is bound to proteins [7]. Iron is present in enzymes as well as in cellular structures such as DNA and mitochondria. Iron also gives hemoglobin the ability to carry oxygen [8].

The human body contains between 3–4 g of iron. Daily iron-dependent metabolic processes are possible because iron is recycled, released from hepatic stores, or absorbed [9]. Daily deficits arise that cannot be met by intake alone. Since two thirds of the iron is present in hemoglobin, which has a lifespan of 120 days, the body relies heavily on this source [10]. With ageing, the erythrocyte membrane expresses neoantigens. In the spleen, macrophages dismantle the heme and release the iron [11]. Although the enteral source of iron accounts for only 0.1% of daily requirements, this source cannot be disregarded [12].

In the duodenum, FeII and FeIII are taken up into the enterocyte through the apical membrane (Figure 2). The former takes the path of the divalent metal transporter-1 (DMT1) and the later uses the mobilferrin integrin pathway (IMP) [13]. DMT1 has a number of isoforms that occupy different cells and different organelles within the same cell. Similar to the duodenal enterocyte, erythrocytes also take up iron through DMT1, although this is of a different isoform [14]. For IMP to transport FeIII, pairing with mobilferrin or beta 3 integrin proteins must take place. While IMP is specific for iron, DMT1 also allows for copper, zinc, and manganese absorption [15].

The bulk of the absorbed inorganic iron is in the FeII form. FeIII from the diet undergoes reduction. This reaction is catalyzed by duodenal cytochrome B (cytB) at the level of the luminal, brushlike border of the duodenal enterocyte [17]. Vitamin C is an essential electron donor involved in iron absorption by acting as a reducing agent. Heme from meat sources account for two thirds of the absorbed iron and involve a heme carrier protein. Following endocytosis, ferrous iron is freed by the heme oxygenase (HO) [17]. Once in the enterocyte, FeII is either stored in the form of ferritin, taken up by mitochondria, or exported through ferroportin (FPT) in the basolateral membrane into capillaries. Similarly, macrophages engulf senescent hemoglobin, which are broken down, making heme available for HO in the cytoplasm to liberate iron from recycling. Iron is either coupled with ferritin for storage or, more commonly, exported [18]. In both cells, FPT is the sole iron exporter.

FPT is responsible for releasing iron into the plasma from all the different sources. Located on the membranes of different cells—for example, enterocytes, hepatocytes, and macrophages—FPT requires ferroxidases for iron export. After FeII is exported, hephaestin (HEPH) and ceruloplasmin (CP) convert FeII to FeIII at the enterocyte while ceruloplasmin oxidases iron at the level of macrophages and hepatocytes. Experimental models in which ferroxidases were knocked out resulted in decreased plasma iron levels with iron stores being overloaded. This shows how important the proper functioning of FPT really is for iron metabolism [18,19].

Plasma FeIII can assume two forms, non-transferrin-bound iron (NTBI) or transferrin-bound iron (TBI). Although not a major player, NTBI can be taken up by some tissues even in the absence of transferrin. This was evident in experimental models where transferrin-knocked-out mice still developed iron overload in nonhematopoietic organs [18]. An important site of iron use is the bone marrow (BM), where hematopoiesis takes place. At the target organ, transferrin is internalized together with transferrin receptor 1 (TfR1). After releasing the iron, TfR1 and the carrier protein reassume the initial role [20].

The importance of TfR1 in hematopoiesis was demonstrated in an experimental model, where TfR1-knock-out mice displayed decreased iron content and impaired iron homeostasis in hematopoietic stem cell [21]. The significance of this receptor is not limited to hematopoiesis alone. In response to hypoxia, an overexpression of TfR1 can be seen, where hypoxia-induced transcription factor 1-a (HIF-1a) upregulates both DMT1 and TfR1 [22,23]. Overexpression of TfR1 can also be detrimental, as seen in tumor cells, due to accumulating iron, which generates ROS through the Fenton reaction [24].

Iron stores are polycompartmental with enterocytes, macrophages, hepatocytes, and plasma representing, to different degrees, sites of storage. FeII is oxidized to curb the redox potential and binds to ferritin for storage. After being absorbed through the apical membrane of the enterocyte, iron molecules are carried by cytoplasmic proteins called chaperones, namely, poly (rC)-binding protein 1 (PCBP1). Ferritin H, an isoform of ferritin, represents the main form of iron storage at this level. Although not the largest deposit, this site cannot be neglected. Experimental deletion of ferritin H leads to an accumulation of iron in other organs with harmful effects. In the plasma, iron is kept bound to the light isoform of ferritin. Ferritin L mirrors the amounts of iron in stores, which is valuable in medical practice [18].

### 2.2. Iron Homeostasis

Control over the iron stores is achieved by feedback loops involving numerous pathways. A key player of these mechanisms is hepcidin, a protein that blocks the FPT (Figure 3). How hepcidin is regulated, which triggers or suppresses production, makes up a complex web of mechanisms that are active both in health and in illness. Unfortunately, many of the proposed pathways are not yet known or are insufficiently explained.

Iron levels begin a feedback loop in which TfRs are passed through to be up- or downregulated. There are two TfR isoforms, namely, TfR1 and TfR2. The former is part of a feedback loop involving cellular iron stores and is present throughout the body. The latter is only present on hepatic cells. While TfR1 is upregulated by depleted iron stores and hypoxia, TfR2 displays a proportional relationship with levels of circulating iron-transferrin complexes. Herbison et al. demonstrated on HuH7 liver cell cultures that while iron status influences TfR expression, silencing both isoforms does not significantly influence iron uptake. This finding indicates the existence of other key components [26].

Intracellular iron is also recognized by bone morphogenic protein (BMP). Increased iron levels are met by hepatic production of BMP6 and BMP2, which are necessary for the signaling of hepcidin production. Once synthesized, BMPs bind to their respective receptors. As a result, downstream phosphorylation of SMAD1/5/8 takes place. This step was proven to be dependent on MyD88, a gene expressed in both macrophages and hepatocytes. The phosphorylated SMAD1/5/8 forms a complex with SMAD4 and, together, they have a stimulatory effect on hepcidin synthesis in the nucleus [27].

The microcosm of iron homeostasis is achieved by means of FPT genes, which encode similar proteins but with distinct roles. FPT 1A differs from FPT 1B through the 5′ iron-responsive element (IRE). The former possesses the 5′IRE, which functions as a suppressor of FPT. When intracellular iron levels are low, FPN 1A is expressed to preserve stores. Interestingly, enterocytes and red blood cell precursors express the FPN 1B isoform indicating that iron from these sources is always made available when plasma levels are low [18].

Another feedback mechanism component is represented by the HFE, which is a major histocompatibility complex class I (MHC) protein. Mutations of the gene that encodes HFE, such as those in some forms of hereditary hemochromatosis (HH), denature the structural conformation of the protein. As a result, iron overload and organ damage occur [20]. Studying HH helped shed light on the role HFE has in iron homeostasis and, more exactly, on hepcidin regulation. Experimental mice studies have emphasized that the main role of HFE lies in the liver. This moves away from the initially supposed mechanisms in which hepcidin was regulated by iron levels in macrophages and enterocytes. Thus, the center of hepcidin regulation and also production is the hepatocyte. Key players of the mechanism are the TfR1, TfR2, BMP, BMP receptor (BMPR), and hemojuvelin (HJV). Centered around HFE, these are upstream regulators of hepcidin. Transferrin-bound iron interacts with TfR1, which in resting state, is associated with HFE. As a result of iron-TfR1 interaction, HFE is freed. HFE then links to TfR2 and forms a complex, which includes HJV, BMP, and BMPR; together, they signal the production of hepcidin in an attempt to curb the increasing levels of iron. Hepcidin closes the feedback loop by blocking the FPT and effectively lowers iron levels [16].

The reverse is also true in low iron states, where hepcidin signaling does not take place. On one hand, HFE remains bound to TfR1 and, on the other, matriptase 2 (MT2) catalyzes the HJV at the hepatocyte membrane. MT2 is coded by the TMPRSS6 and is inversely proportional to hepcidin [28]. Other negative feedback mechanisms are triggered in situations where iron demands are high—for example, erythropoiesis, where precursor cells suppress hepcidin expression through erythroferrone [29].

Hepcidin regulation can also occur independently of iron stores. Although the upstream regulators of hepcidin are variable and plenty in number, most converge on two, namely, the SMAD 1/5/8 and STAT3 pathways [30]. Inflammatory conditions, whether chronic or acute, are associated with the production of different cytokines. Interleukin 6 (IL6) is a major inducer of hepcidin and was demonstrated by spiking urinary levels of hepcidin excretion in healthy humans, with a decrease in plasma iron levels in response to exogenous IL6 [31]. Besides the notable IL6, other cytokines are involved in hepcidin induction as well. While IL22 plays a minor role, IL1 is thought to be responsible for enhancing IL6 production and also that of activin B [29].

Activin B produces hepcidin upregulation [25]. Unlike IL6, which takes the STAT3 pathway, inflammation-induced activin B activates SMAD1/5/8 when binding to the BMP receptor. As a member of the transforming growth factor beta family, activin B is a cytokine that uses the type I bone morphogenic protein receptors (BMPRs) ALK2 and ALK3. This shows how pluripotent inflammation really is in inducing hepcidin [32]. The large variety of cytokines produced is a challenge to overcome when developing treatment options. Although hepcidin signaling converges on either SMAD and STAT pathways, studies bring to light the existence of a SMAD and STAT interaction, adding another hurdle in the attempt to stop the hepcidin production chain [33].

## 3. Copper and Zinc Metabolism

### 3.1. Copper

Cu homeostasis is balanced by absorption from the intestinal tract and by excretion in the bile. Neurotransmitters, oxygen delivery, and genetic processes rely on this element. At the enteric level, specific copper transporter 1 (CTR1) and the nonspecific DMT1 absorb copper from ingested diet (Figure 4). Similar to iron, copper relies on reductases such as cytB and Six Trans membrane Epithelial Antigen of The Prostate (STEAP) for the absorption to take place. Inside the cell, chaperone proteins interact with Cu and are either stored or delivered to intracellular organelles. Cytochrome C (cytC) oxidase copper chaperone is responsible for supply to the mitochondria. Furthermore, copper can enter the interstitium by means of copper-transporting AT Pase 1 (ATP7A) transporter at the level of the basal membrane, where it undergoes oxidation by ceruloplasmin. Oxidized copper binds to proteins such as albumin and a2-macroglobulin for transport [34].

ATP7A and copper-transporting AT Pase 2 (ATP7B) are involved in both copper-based enzyme production and copper transport. The former is found at the level of the enterocyte and is responsible for copper absorption, while the ATP7B is responsible for excretion into the bile. In copper-depleted states, the lack of cuproenzyme production translates into iron deficiency as iron requires HEPH and CP for absorption [6].

In conditions of iron depletion, copper is absorbed mainly through the DMT1. This is thought to occur due to an increased expression of the iron transporter at the brush border of the enterocyte [35]. The role of copper in iron deficiency is to stimulate iron mobilization from stores by producing ceruloplasmin synthesis.

When iron is abundant, such as in hemochromatosis, ceruloplasmin levels are low and associated with copper deficiency. Similarly, here, iron supplementation was associated with decreased copper levels, which may be of importance for certain groups of patients receiving iron treatment, e.g., pregnant woman [34].

### 3.2. Zinc

Unlike the other divalent metals, Zn does not require reduction or oxidation for transmembrane transport. In the divalent form, Zn is absorbed through the Zrt-, Irt-related protein (ZIP) from the intestinal lumen. From the many isoforms present, ZIP4 is found at the brush border of the enterocyte and is most relevant for uptake. Zinc transporter 1 (ZNT1) is another membrane protein located at the basolateral membrane, which is responsible for transferring zinc into the blood. While intracellular trafficking is thought to rely on metallothionein (MTH) and zinc chaperones (ZnCH), the exact mechanisms and proteins are not yet known [36].

## 4. Medical Uses of Iron, Copper, and Zinc Nanoparticles

Understanding the metabolism of these ions in normal and abnormal conditions allows for the use of nanoparticles in treatment and diagnosis (Figure 5). NPs can be used to alter the levels of the ions already present in organisms or, when produced as ion-NPs, to take advantage of homeostatic pathways present in the human body. Nanoparticles are produced by physicochemical processes, which use heat, ultrasound, or microwaves. Various geometrical shapes and sizes can be obtained by altering the reactants’ concentrations or the conditions of the reaction. When arranged in more complex structures, nanoparticles gain superior properties such as improved bioavailability, targeting capabilities, and theragnostic properties [37]. Spherical and octopod particles, apart from their imaging properties, also possess the ability to produce heat [38]. Size also influences their properties. One-pot synthetic procedure, as described by Lu Z. et al. and later modified by Ledda M. et al., has been used to produce ultra-small particles. Ultra-small iron oxide nanoparticles (USIONPs), due to their size of less than 5 nm, penetrate cells more readily while retaining their paramagnetic properties. Moreover, increasing evidence supports vascular as well as blood homeostasis applications of iron oxide nanoparticles, including thrombolysis, vascular grafts and stents, atherosclerosis treatment, or cardiovascular regeneration [39].

### 4.1. Nanoparticles to Improve Bioavailability

#### Iron Administration

Iron deficiency causes anemia with many deleterious systemic effects. Insufficient intake is an issue most prevalent in underdeveloped countries and in cohorts of western countries when a self-restricted diet is adopted. Correcting iron deficits is especially important in surgical patients whereby reversal of iron deficiency lowers transfusion requirements [40]. This conduct represents the first pillar of all patient blood management programs [41,42]. Iron can be administered enterally or intravenously, with the former being most common [43]. Food fortification is an option to prevent and treat iron deficiency; however, fortification is limited by bioavailability and side effects. When used for fortification, iron alters the taste of food [44]. Notable side effects of oral iron medication are nausea, vomiting, and diarrhea.

Iron NPs have been produced to deliver iron in an elemental form, with improved properties. Absorption occurs in the duodenum by alternate mechanisms to those of DMT1 and duodenal cytochrome B (DCYTB). NPs adhere to the cellular membrane and subsequently undergo endocytosis [45]. This was demonstrated in vitro on cellular lines and rat experimental models, where silencing the relevant genes of the classic iron absorption pathway did not inhibit the uptake of iron nanoparticles [46]. Nevertheless, DMT1 is essential for overall iron uptake, as it is believed to be the transporter involved in freeing the iron from lysosomes following endocytosis [47].

Iron sulfate is the pioneer of NPs and is followed by newer particles [48]. Negatively charged or liposoluble particles cross the membrane more easily while mesoporous iron has higher bioavailability compared with iron salts due to the smaller size and larger surface area after dissolution at gastric pH [49,50].

To improve NPs, different molecules are used to coat the particles. Food fortification is possible using protein-coated iron NPs to increase solubility and absorption. Coating with either albumin or the milk protein’s amyloid fibril component improves hydro solubility and prevents the aggregation and precipitation of iron NPs at gastric pH. This allows for enzymatic degradation without aggregation [44,51]. Alternatively, the synthetic ferritin FeIII oxo-hydroxide nanoparticles also demonstrate good bioavailability with few side effects when compared with the ferrous iron fortifiers [52,53].

Encapsulation of iron oxide magnetic NP in liposomes is also efficient at increasing bioavailability and reducing toxicity. This method of drug delivery prevents immune-mediated hemolysis and, compared with uncoated particles, causes fewer side effects. In terms of efficacy, liposomal encapsulation produces significant increases in hemoglobin and red blood cell levels compared with both uncoated NP and iron sulfate [54]. Similarly, exosomes that are composed of an outer bilayer membrane, grant the NP resistance against biodegradation and also the ability to cross cellular membranes with hydrophilic contents. Other benefits of using exosomes lie in the ability to adjust the delivered dose and target specific cell receptors [55].

Biocoating is not limited to human molecules. *Lactobacillus fermentum* is a commensal organism used for biocoating of NPs to enhance the resistance against gastric acid. In experimental models, iron absorption was confirmed by magnetic resonance imaging and relevant changes in biological markers. Increased hemoglobin and decreased DCYBT, DMT1, and hepcidin levels were present [56].

The intravenous form of iron oxide NP is used in cases where the oral route is impracticable or inefficient. Following administration, the drug is trapped in the endoplasmic reticulum, where iron is released for uptake by macrophages or binds to iron carrier proteins [57]. Parenteral iron needs coating to prevent the generation of ROS. This is achieved by coating iron oxide or hydroxide NPs with dextran, sucrose, carboxymaltose, ferumoxytol, isomaltoside, or other nonbiological compounds. Apart from the potential redox reactions generated, iron may promote bacterial growth [58].

Approved for coating are monosaccharides, which further improve biocompatibility and are successfully used intravenously for the treatment of iron deficiency [56,59]. Joined with iron sulfate, the lipid nanoparticle compritol 888 ATO initially behaves similar to a quick release drug and later displays sustained release. The resulting formula achieves higher plasmatic levels compared with conventional iron supplements [60].

### 4.2. Targeting Capabilities

NPs can be designed to behave in specific ways depending on their coating, method of synthesis, or morphology. Biological molecules give the NPs cell-targeting capabilities [61] via specific receptor pathways. This property is employed in treatment of different human diseases [55]. For example, hemin, which is a ferric iron containing protoporphyrin, increases the absorption of iron NPs through the heme-specific transport [62], while transferrin-bound NPs interact with the transferrin receptors on different cells and tissues. The nanoparticle and receptor complexes are endocytosed followed by the release of the NP’s cargo [63]. Targeting cells and tissues open new frontiers for treatment and diagnostics.

#### 4.2.1. Iron Chelation

Targeted gene silencing to reduce iron absorption can be used in the treatment of HH. The disease is characterized by iron overload caused by excessive intestinal absorption of iron and has severe systemic implications. Silencing RNA (siRNA) aimed at the gene responsible for the expression of DMT1 can be transfected via gelatin NPs. This multicompartmental particle demonstrated increased specificity for the duodenal enterocytes, sparing other iron storage organs. Coating the gelatin NP with Eudragit polymer further improved drug stability [64]. Another way to deliver siRNA is by lipid-based NPs that are extracted from ginger and similar to exosomes. This naturally occurring compound is less toxic and produces less inflammation compared with synthetic alternatives. Although this method of reducing iron absorption seems promising, humans possess the ability to absorb heme-associated iron through mechanisms independent of DMT1. Since most of the studies have been conducted on murine models that do not possess the capability to absorb heme and the side effects of iron chelators used for the treatment of HH are poorly tolerated by patients, it can be concluded that more studies are required on this issue [65].

Targeted gene silencing is also adopted in the treatment of thalassemia. This disease is characterized by a continuous anemic state, which is associated with decreased levels of hepcidin. The resulting iron overload can be addressed by targeting these downstream components of iron metabolism. Blocking the expression of hepcidin’s protease matriptase 2, by silencing the TMPRSS6 gene, results in higher levels of hepcidin, which in turn reduces the absorption of enteral iron. In a murine model experiment, the authors have successfully prevented iron overload. To deliver the genetic load, lipid NP containing antisense oligonucleotides were used [66].

Hepcidin targeting is particularly efficient in HH and thalassemia [67]. Extrapolating from this idea, future studies could investigate the effects of silencing hepcidin expression in inflammatory anemia, which is a major issue in the intensive care unit (ICU). Increased hepcidin is associated with poor outcome and increased mortality in this population, with iron dysmetabolism and inflammation [68,69]. The polypeptide synthesis is stimulated by proinflammatory mediators and causes functional iron overload with secondary anemia.

#### 4.2.2. Antimicrobial Treatment

The complex immune response whereby metal ions are tightly controlled in an attempt to fight invading organisms is referred to as nutritional immunity. Iron, copper, and zinc play essential roles during systemic inflammatory response syndrome. Levels of iron and zinc are kept low by the host organism to limit bacterial proliferation; the latter thrives in conditions of iron and zinc overload. Bacterial growth occurs when iron is highly available, often seen after transfusions. The same is also true in genetic disorders with elevated iron [70]. Targeted manipulation of ion levels represents a new therapeutic approach in sepsis.

Thanks to their properties, carrier nanoparticle and metal oxide complexes can be used as chemotherapeutic agents for targeting microbes. Sepsis is a leading cause of death, especially in intensive care units. Over the last century, numerous antibiotics have been developed due to the increasing bacterial resistance. Divalent metal NPs have been used as antimicrobial agents and for coating of medical devices. Coatings on devices prevent the formation of biofilm produced by different drug-resistant bacteria. In an experimental model, using a sol-gel process, iron-doped copper NPs successfully inhibited biofilm formation [71,72].

Divalent elements are catalysts of Fenton reaction producing reactive oxygen species [73]. Based on the premise that both iron and copper produce ROS, NPs have been loaded and aimed against microbes, viruses and fungi [74,75]. Bacteria internalize the nanoparticle complexes that lead to cellular death. Another mechanism is by disruption of DNA, which halts the cellular processes taking place. Such interferences prevent the invading organisms from dividing with some sparing effects on the host [76]. Evidence of copper NP supplementation revealed by animal experiments in vivo demonstrates the antioxidant effect on blood with enhanced catalase (CAT) activity resulting in hydrogen-peroxide-optimized detoxification [77].

Both the levels of ions and the microenvironments in which they play a role are essential. The antimicrobial effect is not limited to the ions but is also conferred by their ligands. One such ligand is the naturally occurring chitosan found in crustaceans [78]. When comparing ligands alone with ligand and metal oxide complexes, the nanosized complexes demonstrate superior efficacy against both bacteria and fungi. Data supporting this showed that Zinc-Cobalt ferrite NPs are efficient against *Klebsiella pneumoniae* while Copper-Cobalt ferrite NPs have antifungal effects [79]. It is worth mentioning that in comparison with existing standard therapy, these NPs have not been superior [80] and represent an area of interest for future research. Nevertheless, they may be used as an alternative drug therapy in some drug-resistant microbes.

Targeted delivery and control with NPs is again evident with zinc. While Zn deficiency is detrimental for the host, starvation of bacteria has beneficial effects, as seen in experimental models where Zn chelation re-established antibiotic susceptibility in carbapenem-resistant microorganisms. Supplementing with zinc in an attempt to boost immunity of the host is controversial. It is true that zinc shortens the duration of some viral respiratory tract infections, probably by boosting the immune system; however, the same beneficial effect has not been seen in critically ill patients. Furthermore, when zinc is administered in high doses to support the immune response, it competes with copper for metallothionein, resulting in copper deficiency. High doses result in toxicity, as seen with developing mice experimental models [81].

#### 4.2.3. Biological Sample Analysis

A thorough understanding of ion physiology, nanomaterials, and their interactions allowed for the creation of polymers that act as nanoparticle receptors, to determine the titres of specific nanosized molecules, including proteins. As a result of reduced costs of production, high versatility, specificity, and sensitivity, nanosized molecularly imprinted polymer plates are of interest.

Iron status biomarkers are used to predict response to iron therapy and blood transfusions [82]. Some markers have been tested to predict response to iron treatment; however, all presented shortcomings such as reference value discordance between manufacturers and published data [31]. Conventional tests lack clear cut-off values and must be validated against a gold standard in patients with significant inflammation [83].

Hepcidin-specific plates have been developed and tested with success to determine sample levels with ranges as low as 1 nM and may be quantified in as little as three minutes [84,85]. This technology may prove useful at the point of care in ICU, where marked inflammation is present in critically ill patients. As a result, hepcidin is upregulated and results in the sequestration of iron [25]. Hepcidin proved useful in detecting iron deficiency, even in cases of inflammation. Low levels indicate iron depletion while high levels show inflammatory anemia. In ICU patients who initially presented inflammation but later developed iron deficiency, hepcidin followed a decreasing trend as iron loss progressed [86]. These patients may benefit from iron supplements to treat anemia. Likewise, ferritin can also be determined with great accuracy. Quantum dots increase the sensitivity of the Western blot technique and yield a 20-fold increase in sensitivity for human ferritin when used together [87].

Urine samples are accessible and contain small proteins that serve as biomarkers used in sepsis and cancer screening. Expanding on treatment options is crucial in sepsis; however, often the culprit is difficult and timely to identify. Bacteria that metabolize Cu can be rapidly identified and targeted in biological products using fluorescent spectroscopy. While *E. coli* can be detected in urine using iron quantum clusters by interacting with Cu from the bacteria, Zinc-Cobalt ferrite nanoparticles may be used as an antimicrobial agent [79]. Establishing the microorganism causing sepsis is often a race against time and can prove invaluable in critically ill patients [88]. In oncology, magnetic Concanavalin A-NPs have been used to detect Cathepsin C and transferrin, which may serve as biomarkers in the urine of small-cell lung cancer patients [89].

Furthermore, synthesis of dealloyed nanoporous gold (NPG)/ultrathin CuO film nanohybrid has been reported as an efficient detection method of glucose, therefore becoming a promising tool for glucose tests [90].

### 4.3. Theranostics

By understanding the physiology of iron, copper, and zinc metabolism, and by taking advantage of the pathophysiological changes in diseases such as neoplasms, NPs are concomitantly used for treatment and diagnostics. This concept is termed theragnostics. USIONPs, due to their paramagnetic properties, and when exposed to an external magnetic field, specifically accumulate in intended cells [91]. Once in the tissue, the NPs may take up the role of contrast agent, chemotherapeutic agent, or drug mule. The latter provides better bio availability for the cancer treatment with reduced toxicity. The cargo of a drug carrying NP may be released or activated using photo thermal or magnetic fields [39]. Concomitantly using magnetic resonance imaging in a theragnostic approach provides personalized treatment in the field of oncology. Small nanoparticles such as octopods, when exposed to a magnetic field, release heat energy. This is the basis for localized hyperthermia. This method allows for precise targeting of neoplastic tissues while preventing any damage from occurring to healthy cells [37,38].

Iron oxides have the ability to act as contrast agents [57,92]. Paramagnetic properties are given by the iron core as well as the shells of the NP—for example, polydopamine, gold, or quantum dot coatings [59]. An alternative to synthetic contrast agent is ferritin H with ferrous iron-loaded core—termed magneto ferritin, it is used as magnetic resonance imaging (MRI) contrast [93]. Further labelling the outer core with radioactive iodine, the newly formed contrast agent is not only magnetic signaling but also positron emitting. This hybrid contrast agent can be used for combined positron emmiting tomography (PET)-computer tomography (CT) or MRI imaging. Due to the specific uptake of ferritin, namely, by transferrin receptor 1, the contrast agent displays specificity for cells with increased numbers of receptors, such as cancer cells [94]. This transport pathway not only allows for specific uptake but also solves the problem of multiple injections for this method of imaging [95]. Labelling cells and tissues is not limited to MRI. Gold-labeled transferrin is visualized using confocal microscopy, with no cytotoxic effects. The labeled transferrin maintains the ability to recognize and interact with the specific receptor on cell surface, allowing for the visualization of transferrin-receptor-rich cells, with an increased iron metabolism [96].

NPs present good penetrability, even crossing the blood-brain barrier [97], and target tumor cells, which express high numbers of transferrin receptors [98]. Transferrin-bound nanoparticles are adopted in theragnostics [99] to achieve improved results. Transferrin is loaded with metallofullerene to inhibit the much-needed iron uptake by the neoplastic cells [100]. Moreover, copper oxide nanoparticles have been demonstrated to efficiently target the tumor-initiating cells in pancreatic adenocarcinoma both in vitro and in vivo. The effect was also linked to reverse mitochondrial membrane polarity and excessive ROS production, resulting in enhanced apoptosis in the tumor-initiating cells [101]. Several Cu(I) complexes have been designed in a recent study. Among them, several prototypes have demonstrated ROS-generation abilities and cellular apoptosis-necrosis disequilibrium in prostate cancer cells [102]. Even more data exist regarding the potential of zinc oxide nanoparticles as anticancer agents. Interestingly, mechanisms such as p53 suppression, bax upregulation, BCL-2 silencing, and DNA fragmentation were documented to appear predominantly in cancer cells and less in normal cell lines, suggesting an intrinsic selectivity in the antitumor effect of zinc oxide nanoparticles [103]. Although the precise mechanism responsible for the onset of intrinsic selectivity of divalent ion nanoparticles is still under intensive research, the effect was linked to catalytic activity through Fenton-like reactions and was reported for several tumors, such as human ovarian cancer [104], non-small-cell lung cancer [105], or breast cancer [106].

## 5. Conclusions

Metabolism of divalent metal ions presents a high level of complexity and multiple interactions with several key physiological systems. Understanding the physiology and physiopathology of ion metabolism allows for nanoscale manipulation of signaling mechanisms, effects, or end-products. It is evident that nanoparticles possess pleiotropic properties used in the diagnostics and treatment of different diseases. Of note is the theranostic application in cancer treatment, which is a growing field of interest. Nonetheless, sepsis and anemia are still major causes of morbidity and mortality worldwide and require further research.

## Figures and Tables

**Figure 1 nanomaterials-11-02958-f001:**
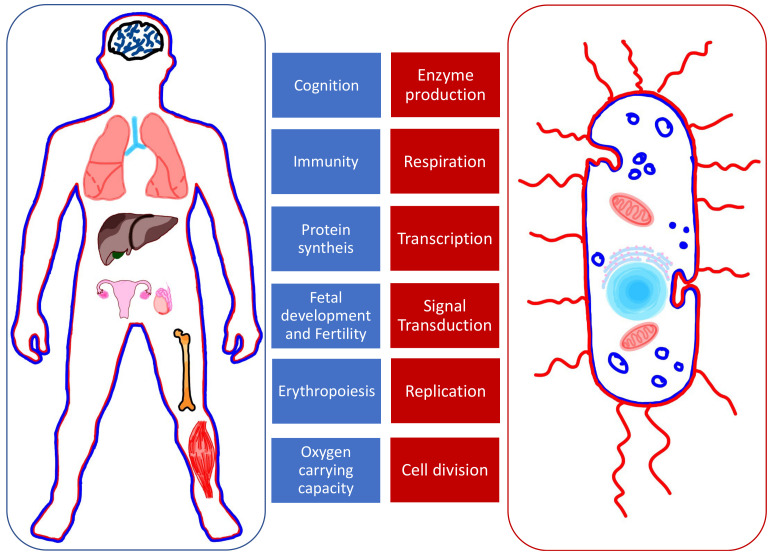
The roles of Fe, Cu, and Zn in human organs (**left, in blue**) and bacteria (**right, in red**).

**Figure 2 nanomaterials-11-02958-f002:**
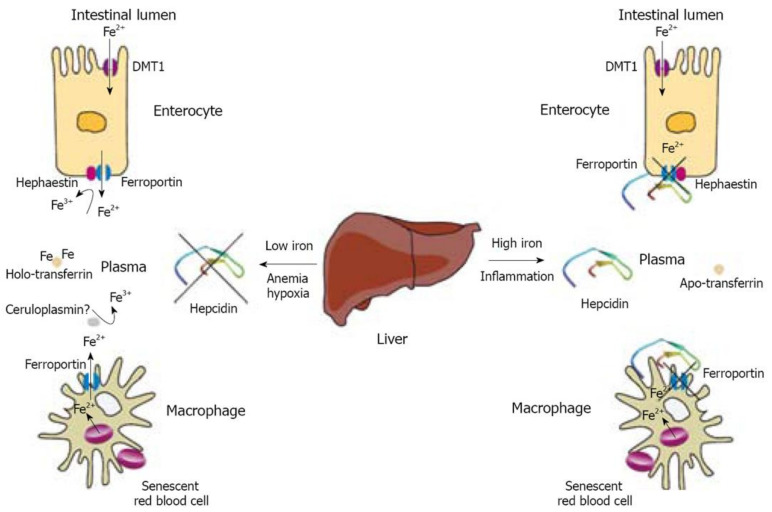
Regulation of iron efflux from enterocytes and macrophages by hepcidin. Reproduced with permission after Pantopoulos K. et al., 2008 [16].

**Figure 3 nanomaterials-11-02958-f003:**
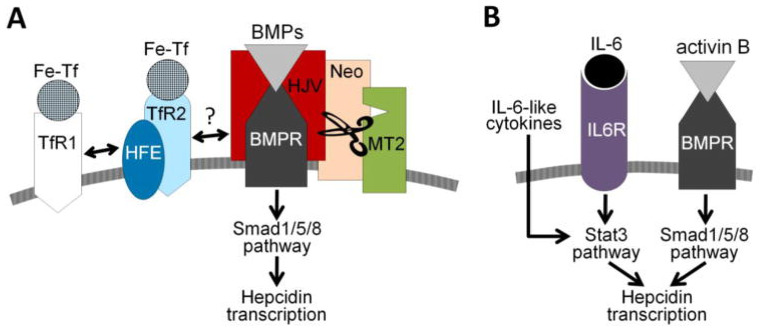
Regulation of hepcidin transcription. (**A**) Hepcidin regulation by the extracellular iron. (**B**) Hepcidin regulation by inflammation. Reproduced with permission after Ruchala et al., 2014 [25].

**Figure 4 nanomaterials-11-02958-f004:**
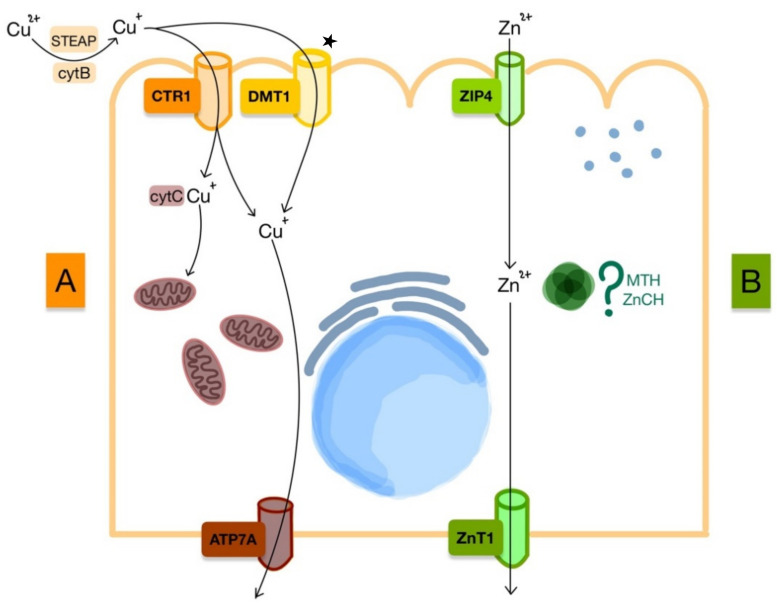
Absorption of ions by enterocyte. (**A**) Cu absorption during normal homeostasis and in iron deficiency; main pathway during iron deficiency marked with ^★^; (**B**) Zn absorption during normal homeostasis.

**Figure 5 nanomaterials-11-02958-f005:**
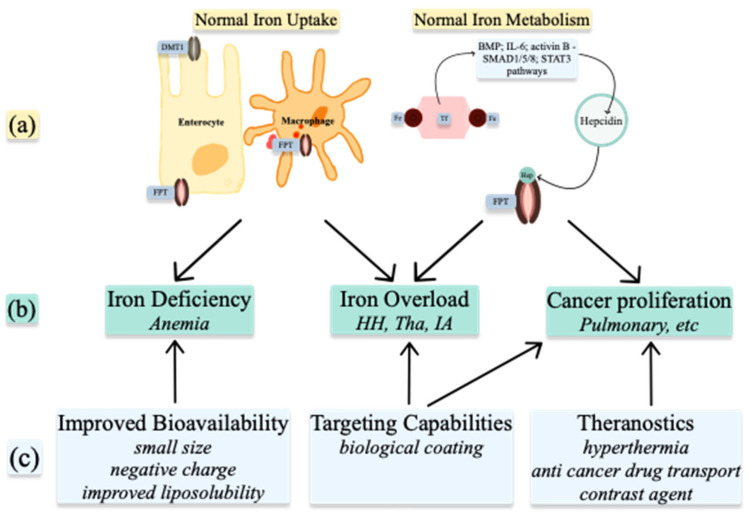
Medical uses of Iron nanoparticles: (**a**) normal Iron homeostasis; (**b**) pathologies resulting from abnormal Iron homeostasis; (**c**) properties of nanoparticles. DMT1—divalent metal transporter 1; FPT—ferroportin; Fe—iron; Hep—hepcidin; BMP—bone morphogenic protein; IL-6—interleukin 6; HH—hemochromatosis; Tha—Thalassemia; IA—inflammatory anemia.

**Table 1 nanomaterials-11-02958-t001:** Common disorders secondary to Fe, Cu, Zn deficiency or overload.

Disorder	Implicated Divalent Ion	Levels on Ion
Hemochromatosis	Iron	↑
Thalassemia	Iron	↑
Inflammatory anemia	Iron	↑
Iron-deficient anemia	Iron	↓
Copper-deficient anemia	Copper	↓
Menkes syndrome	Copper	↓
Wilson disease	Copper	↑
Immunocompromise	Zinc	-

## Data Availability

Not applicable.

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
