# Peer review of "Iron, Copper, and Zinc Homeostasis: Physiology, Physiopathology, and Nanomediated Applications"

_nanomaterials, 2021, doi:10.3390/nano11112958_

Round 1

Reviewer 1 Report

In principal, the authors present a well-written review on iron, copper and zinc homeostasis, physiology, physiopathology and nanomediated applications.

However, the reviewer thinks that the topic is not appropriate to the readership of Nanomaterials. The part on nanomaterial applications is relatively very short and superficial and doesn’t bring new detailed information to the reader.

Therefore, the reviewer doesn’t recommend the manuscript for publication in Nanomaterials.

Author Response

We thank the reviewer for the expert opinion formulated. We have reformulated and enriched the nanomaterial part to better fit the journal topic.

Reviewer 2 Report

Really nice and sell written overview. The review focuses on the mechanisms involved in Fe, Cu and Zn homeostasis and in the role of nanotechnology for diagnosis and treatment of these metal irons disorders. Nice work.

Author Response

We thank the reviewer for the encouraging words and appreciation, as well as for the time and effort spent reviewing the paper.

Reviewer 3 Report

The review provides an overview of the mechanisms involved in the homeostasis and pathophysiology of Fe, Cu and Zn. It also aims to describe the state of the art on the role of nanoparticles in diagnostics and therapy. The review will be of interest to nanotechnology researchers by adding value to the current knowledge base. However, some important points need to be addressed.

The review is not enough complete. In section 3 the authors describe the different therapeutic uses of nanoparticles, it seems to me that the title "The role of nanoparticles in iron, copper, and zinc metabolism" does not correspond to the actual content. Furthermore, the authors investigate the aspects concerning iron nanoparticles and the other metals very little. In the context of theragnostic applications it would be very interesting to describe the use of nanoparticles for the delivery of cells / drugs and for iron-based hyperthermia. Possible references are:

  1. Xie , Z. Guo , F. Gao , Q. Gao , D. Wang , B. S. Liaw , Q. Cai , X. Sun , X. Wang and L. Zhao , Theranostics, 2018, 8 , 3284 —3307
  2. Ledda, D. Fioretti, M. G. Lolli, M. Papi et all. Nanoscale, 2020, 12, 1759

Cotin G, Blanco-Andujar C, Perton F, Asín L, at al.. Nanoscale. 2021 Sep 2;13(34):14552-14571

Friedrich RP, Cicha I, Alexiou C. Nanomaterials . 2021 Sep 8;11(9):2337

The conclusions are too concise and vague. It would be appropriate to explain them better

Author Response

Point 1: The review is not enough complete. In section 3 the authors describe the different therapeutic uses of nanoparticles, it seems to me that the title "The role of nanoparticles in iron, copper, and zinc metabolism" does not correspond to the actual content.

Response 1: Very appropriate observation, thank you. We have modified the title to better fit the actual content. Moreover, several novel paragraphs have been added, as to include more information regarding not only Iron , but also Copper and Zinc-content nanoparticles and their applications. A total of 19 new references have been added to the text for improvement, as requested.

Point 2: Furthermore, the authors investigate the aspects concerning iron nanoparticles and the other metals very little. In the context of theragnostic applications it would be very interesting to describe the use of nanoparticles for the delivery of cells / drugs and for iron-based hyperthermia. Possible references are:

  1. Xie , Z. Guo , F. Gao , Q. Gao , D. Wang , B. S. Liaw , Q. Cai , X. Sun , X. Wang and L. Zhao , Theranostics, 2018, 8 , 3284 —3307
  2. Ledda, D. Fioretti, M. G. Lolli, M. Papi et all. Nanoscale, 2020, 12, 1759
  3. Cotin G, Blanco-Andujar C, Perton F, Asín L, at al.. Nanoscale. 2021 Sep 2;13(34):14552-14571
  4. Friedrich RP, Cicha I, Alexiou C. Nanomaterials . 2021 Sep 8;11(9):2337

Response 2: We agree with the reviewer's suggestions. The indicated valuable reference sources have been inserted in the text, as suggested.

Point 3: The conclusions are too concise and vague. It would be appropriate to explain them better.

Response 3: We agree with the opinion formulated by the reviewer. Conclusions have been reformulated as suggested.

Round 2

Reviewer 3 Report

The manuscript can be published in its current form.

Author Response

We thank the Reviewer for the time and input provided.